# The Role of miR-21 in Osteoblasts–Osteoclasts Coupling In Vitro

**DOI:** 10.3390/cells9020479

**Published:** 2020-02-19

**Authors:** Agnieszka Smieszek, Klaudia Marcinkowska, Ariadna Pielok, Mateusz Sikora, Lukas Valihrach, Krzysztof Marycz

**Affiliations:** 1Department of Experimental Biology, The Faculty of Biology and Animal Science, University of Environmental and Life Sciences, 50-375 Wroclaw, Poland; klaudia.marcinkowska@upwr.edu.pl (K.M.); ariadna.pielok@upwr.edu.pl (A.P.); mateusz.sikora@upwr.edu.pl (M.S.); krzysztof.marycz@upwr.edu.pl (K.M.); 2Laboratory of Gene Expression, Institute of Biotechnology CAS, Biocev, 25250 Vestec, Czech Republic; lukas.valihrach@ibt.cas.cz; 3International Institute of Translational Medicine, Jesionowa 11 St, 55-124 Malin, Poland; 4Collegium Medicum, Cardinal Stefan Wyszyński University (UKSW), Woycickiego 1/3, 01-938 Warsaw, Poland

**Keywords:** miR-21-5p, osteogenesis, differentiation, precursor cells, osteoblasts, osteoclasts

## Abstract

MiR-21 is being gradually more and more recognized as a molecule regulating bone tissue homeostasis. However, its function is not fully understood due to the dual role of miR-21 on bone-forming and bone-resorbing cells. In this study, we investigated the impact of miR-21 inhibition on pre-osteoblastic cells differentiation and paracrine signaling towards pre-osteoclasts using indirect co-culture model of mouse pre-osteoblast (MC3T3) and pre-osteoclast (4B12) cell lines. The inhibition of miR-21 in MC3T3 cells (MC3T3*_inh21_*) modulated expression of genes encoding osteogenic markers including collagen type I (*Coll-1*), osteocalcin (*Ocl*), osteopontin (*Opn*), and runt-related transcription factor 2 (*Runx-2*). Inhibition of miR-21 in osteogenic cultures of MC3T3 also inflected the synthesis of OPN protein which is essential for proper mineralization of extracellular matrix (ECM) and anchoring osteoclasts to the bones. Furthermore, it was shown that in osteoblasts miR-21 regulates expression of factors that are vital for survival of pre-osteoclast, such as receptor activator of nuclear factor κB ligand (RANKL). The pre-osteoclast cultured with MC3T3*_inh21_* cells was characterized by lowered expression of several markers associated with osteoclasts’ differentiation, foremost tartrate-resistant acid phosphatase (*Trap*) but also receptor activator of nuclear factor-κB ligand (*Rank*), cathepsin K (*Ctsk*), carbonic anhydrase II (*CaII*), and matrix metalloproteinase (*Mmp-9*). Collectively, our data indicate that the inhibition of miR-21 in MC3T3 cells impairs the differentiation and ECM mineralization as well as influences paracrine signaling leading to decreased viability of pre-osteoclasts.

## 1. Introduction

MicroRNAs (miRNAs) are a class of non-coding, single stranded RNAs with average length around 22 nucleotides. The molecules are highly conserved across species and have various biological functions that are connected with the regulation of gene expression by targeting 3′-untranslated region (3′-UTR) of messenger RNA [1,2]. MiRNAs regulate multiple processes crucial for homeostasis maintenance, such as cell proliferation, differentiation, survival and apoptosis. Mounting evidence indicates that miRNAs play a vital role during osteogenesis process by modulating expression of genes essential for recruitment and differentiation of various bone cells, including progenitor and mature cells [2,3]. MiRNAs are also important molecules that control the process of extracellular matrix (ECM) mineralization. For instance, miR-29b-dependent suppression of collagen occurs at the late stage of differentiation, which facilitates the maturation of collagen fibrillar matrix for mineral deposition [4]. Furthermore, therapeutic usefulness of miRNAs in terms of bone-associated diseases treatment is widely discussed [5].

MiR-21 represents a molecule that has gained attention as a factor with an important role in bone homeostasis maintenance [1,6,7]. This molecule is predominantly characterized as “oncomiR”, due to the fact that its overexpression was observed in several cancers, including prostate and breast cancer, as well as osteosarcoma [8,9,10]. Most recently, it was shown that miR-21 may promote differentiation of multipotent stromal cells (MSCs) derived from bone-marrow (BMSCs).

The mechanism of miR-21 action is still under debate, nevertheless it was recognized as an important regulator of signaling pathways activated during differentiation of progenitor cells toward osteogenic cells. It has been shown that one of the direct targets of miR-21 is an inhibitory protein Smad7, which modulates signal transduction pathway that is induced by members of transforming growth factor family, i.e., TGF-β and bone morphogenetic proteins (BMPs) [11,12]. A study performed by Li et al. [12] showed that miR-21 deficiency significantly weakened osteogenic potential of BMSCs. Observed effect was associated with decreased expression of master regulator for osteogenesis, i.e., transcription factor *Runx-2* (runt-related transcription factor 2). As a result of miR-21 deficiency, BMSCs showed attenuated osteogenic differentiation and formed poorly mineralized matrix. Moreover, impaired formation of calvarial bone was noted in miR-21-knockout mice [12].

Most recently, it has been demonstrated that miR-21 may promote migration and osteogenic differentiation of BMSCs via PTEN/PI3K/Akt/HIF-1α pathway [7]. MiR-21 exogenously added to BMSCs cultures accelerated the osteogenesis process which resulted not only in increased expression of *Runx-2*, but also secretion of key osteogenic markers such as osteocalcin (OCL), osteopontin (OPN) and BMP-2. Valenti et al. showed that miR-21, derived from sera collected from runners after intense exercise, had an anti-apoptotic effect on BMSCs and promoted their osteogenic potential by targeting PTEN and SMAD7 [13]. Moreover, miR-21 was shown to promote differentiation of MSCs derived not only from bone-marrow. Meng et al. found that miR-21 plays an important role in osteogenic differentiation of umbilical cord blood mesenchymal stem cells (UCBMSCs) via PI3K-AKT-GSK3β pathway and in activating transcription of *Runx-2* [14]. However, there are also contradictory results showing that miR-21 inhibits activation of key osteogenesis regulators. For example, in MSCs derived from periodontal ligament tissue (PDLSCs) overexpression of miR-21 was correlated with decreased expression of alkaline phosphatase (ALP), as well as Runx-2 [6]. The study showed that miR-21 decreased osteogenic potential of PDLSCs by targeting Smad5 molecule, a component of BMPs signaling pathway that is activated during osteoblastogenesis. Furthermore, Wei et al. showed that transfection of cells with miR-21 inhibitor stimulated the osteogenic differentiation of hPDLSCs and improved mineralization of ECM [6].

The role of miR-21 has been studied also in bone resorbing cells (osteoclasts). Suppression of miR-21 was associated with upregulation of osteoclast suppressor programmed cell death protein 4 (PDCD4), and downregulation of osteoclast marker cathepsin K (CTSK) [15]. Thus, miR-21 may be involved in bone biology, not only via promoting mobilization of osteoblast precursors, but also by regulation of osteoclast survival and differentiation [16]. It was also shown that miR-21 knockout mice are characterized by normal skeletal phenotype during development and maintain osteoblastogenesis in vivo. However, miR-21-knockout mice showed increased expression of receptor activator of nuclear factor κB ligand (RANKL) accompanied by decreased level of osteoprotegerin (OPG). Both molecules are major osteoblastic mediators of osteoclastogenesis. RANKL is an essential cytokine promoting differentiation and maturation of osteoclasts, while OPG acts as a decoy receptor for RANKL. OPG inhibits osteoclast differentiation by blocking the interaction between RANKL and RANK, which is a receptor of RANKL [17].

Bearing in mind all this emerging information about the dual function of miR-21 in the process of osteogenesis, we studied the effect of miR-21 inhibition on differentiation of mice pre-osteoblast cell line (MC3T3). Specifically, we analysed the impact of miR-21 down-regulation in MC3T3 cell line (MC3T3*_inh21_*) on their osteogenic potential and paracrine activity towards osteoclasts precursors.

We used indirect co-culture system of MC3T3*_inh21_* and osteoclast precursor cell line 4B12 established by professor Amano’s group [18]. The pre-osteoclastic 4B12 mouse cell line is a model that faithfully recapitulates features of primary osteoclast differentiation showing high expression of c-Fms (macrophage colony-stimulating factor receptor) and RANK (receptor activator of nuclear factor κB) [18,19]. To our best knowledge this is the first study showing the consequences of miR-21 inhibition in osteoblasts on pre-osteoclasts activity. Using the model of indirect co-culture system, we were able to determine the paracrine interplay between MC3T3*_inh21_* and pre-osteoclasts. The analysis included evaluation of matrix mineralization and composition, as well as the analysis of key osteogenic markers expression determined by using reverse transcription quantitative PCR (RT-qPCR), Western blot and immunocytochemical staining.

## 2. Materials and Methods

### 2.1. Pre-osteoblastic Mouse Cell Line MC3T3

MC3T3 cells were cultured in Minimum Essential Media Alpha (MEM-α, Gibco™ Thermo Fisher Scientific, Warsaw, Poland) supplemented with 10% FBS (Fetal Bovine Serum, Sigma Aldrich, Munich, Germany) at constant conditions in incubator at 37 °C, 5% CO_2_ and 95% humidity. Cells were passaged with trypsin solution (StableCell Trypsin, Sigma Aldrich, Munich, Germany). The protocol of MC3T3 detachment included culture washing using Hanks’ Balanced Salt Solution (HBSS) without calcium and magnesium. Following this step, trypsin solution was added to the culture dish in the volume allowing complete coverage of monolayer. The cultures were incubated with the trypsin solution for 5 min at 37 °C in CO_2_ incubator. Detachment of MC3T3 from culture dishes was monitored under inverted microscope with phase contrast (Axio Observer A.1 Zeiss, Oberkochen, Germany). The passage was performed when cultures reached 90% confluence. The MC3T3 used for further experiments were at passage 19 (p19).

### 2.2. Transfection of miR-21 Inhibitor

MC3T3 cells were transfected with miR-21 inhibitor (hsa-miR-21a-5p Anti-miR™ miRNA Inhibitor, Ambition, Thermo Fisher Scientific, Warsaw, Poland) using ESCORT III Transfection reagent (Sigma-Aldrich Sp. zo. o. Poznan, Poland). The transfection reagent was prepared in dilution 1:100, while miR-21 inhibitor was used at concentration equal to 50 nM, which was established based on screening assay results (Appendix A). The reagents were prepared in Minimum Essential Medium Eagle—Alpha Modification (MEM-α). The transfection protocol was followed accordingly to manufacturer’s specification. For the purpose of transfection cells were seeded in 24-well plates at density equal to 20,000 cells per well. Transfection of cells was performed when cultures reached 70% confluence. Cells were used for experiment after 72 h following the transfection. The experiment was divided into two experimental groups: MC3T3 (control) and MC3T3*_inh21_* (miR-21 inhibitor transfected cells).

The effectiveness of miR-21 inhibition was assessed using Two-tailed RT-qPCR. Levels of miR-21 were determined using a protocol described by Androvic et al. [20,21]. Sequences of primers used for the analysis were published previously [22]. Moreover, expression of Runt-related transcription factor 2 was determined as follows: the mRNA levels of Runx-2 were determined by RT-qPCR, while protein expression was detected with Western-blot technique using protocols described below.

### 2.3. Osteogenic Conditions

Osteogenic differentiation of MC3T3 was induced using medium consisted of complete growth medium (MEM-α) supplemented with osteogenic factors—10 nM β-glycerol phosphate disodium salt hydrate (Sigma Aldrich, Munich, Germany) and 50 µg/mL ascorbic acid (Sigma Aldrich, Munich, Germany). The osteogenic medium was changed twice a week, and cultures were maintained in osteogenic medium for 3, 7, and 15 days. After differentiation, the cultures were collected for further analysis.

### 2.4. Co–culture with Pre-osteoclastic Cell Line 4B12

The osteoclast precursor cell line 4B12 used in the experiment was kindly provided by Shigeru Amano from Department of Oral Biology and Tissue Engineering, Meikai University School of Dentistry [18]. The cultures were propagated in complete growth medium (CGM_4B12_) consisting of α-MEM (Sigma Aldrich, Munich, Germany) supplemented with 10% of FBS and 30% of calvaria-derived stromal cell conditioned medium (CSCM). The 4B12 cultures were maintained in incubator at constant conditions (37 °C, 5% CO_2_, and 95% humidity). The cultures of pre-osteoclasts were passaged after reaching confluence of 75%, and then detached from flask by gentle pipetting. The 4B12 used for co-culture experiment were at passage eighteen (p18). Pre-osteoclasts in co-culture system were inoculated at density equal to 3.5 × 10^4^ into upper chamber of 8 µm transwell system (Corning, Biokom, Warsaw, Poland). The cells were maintained in 0.3 mL of CGM_4B12_. Half of the culture medium was changed twice a week. The lower chamber was pre-seeded with MC3T3 pre-osteoblast maintained under osteogenic conditions as indicated in Section 2.3.

### 2.5. Evaluation of Extracellular Matrix Composition

Osteogenic cultures were fixed using 4% paraformaldehyde (PFA) for 15 min at room temperature. Afterwards, cultures were specifically stained with Alizarin Red for calcium deposits detection, and with Safranin O dye for evaluation of proteoglycan content. Staining of extracellular matrix was performed as described previously [23,24]. Obtained specimens were analyzed using inverted microscope Axio Observer A1 (Zeiss, Oberkochen, Germany) and documented with Canon PowerShot digital camera (Woodhatch, UK). The signals obtained after staining were determined using ImageJ and Pixel Counter plugin (version 1.6.0, U. S. National Institutes of Health, Bethesda, MD, USA) as described previously [25,26,27]. The chemical composition of extracellular matrix was evaluated using scanning electron microscope with the energy-dispersive X-ray spectroscopy (SEM-EDX) using the protocol published previously [28].

### 2.6. Detection of Osteogenic Markers Using RT-qPCR

After the experiment, cultures after 3, 7 and 15 days of differentiation were homogenized using 1 mL of Extrazol^®^ (Blirt DNA, Gdansk, Poland). RNA isolation procedure was performed according to manufacturer’s instruction, which is a modified version of the phenol-chloroform method described by Chomczyński and Sacchi [29]. Total RNA was diluted in molecular grade water (Sigma Aldrich, Poznan, Poland). RNA quantity and purity was determined spectrophotometrically at 260 and 280 nm wavelengths (Epoch, Biotek, Bad Friedrichshall, Germany). Total RNA (500 ng) was treated with DNase I using PrecisionDNAse kit (Primerdesign, BLIRT S.A, Gdansk, Poland) before reverse transcription. cDNA synthesis was performed using Tetro cDNA Synthesis Kit (Bioline Reagents Limited, London, UK). Digestion of DNA and cDNA synthesis were performed accordingly to the instructions provided by manufacturer in T100 Thermal Cycler (Bio-Rad, Hercules, CA, USA). RT-qPCR analysis was carried out using the SensiFAST SYBR^®^&Fluorescein Kit (Bioline Reagents Ltd., London, United Kingdom) in CFX Connect Real-Time PCR Detection System (Bio-Rad, Hercules, CA, USA). Each reaction mixture included 1 μL of cDNA in final volume of 10 μL, while primers were used at 0.5 μM concentration. The reactions proceeded according to the following conditions: initial denaturation at 95 °C for 2 min, followed by 45 cycles at 95 °C for 5 s, annealing for 10 s, and elongation at 72 °C for 5 s. The analysis of the dissociation curves was performed to determine the specificity of PCR products. The melting curve was obtained using a gradient from 65 to 95 °C, and the heating rate was 0.2 °C/s. Reaction conditions were described previously [22,27]. Primer sequences are summarized in Appendix A. All reactions were performed in at least three repetitions. The expression of genes was calculated using RQ_MAX_ algorithm and converted into log2 scale as described previously [22]. The transcript levels were normalized to the housekeeping gene – *Gapdh* (glyceraldehyde 3-phosphatedehydrogenase).

### 2.7. Western Blotting Detection of Osteopontin and Runx-2

The cultures were lysed using ice-cold RIPA buffer containing 1% protease and phosphatase inhibitor mix (Sigma Aldrich, Munich, Germany). The concentration of protein was evaluated using the Bicinchoninic Acid Assay Kit (Sigma Aldrich, Munich, Germany). The procedure of the reaction was established previously [22,30]. The concentration of proteins loaded per well was equal to 20 μg. Samples were separated in 12% sodium dodecyl sulphate-polyacrylamide gel by electrophoresis (SDS-PAGE; 100V, 90 min) and transferred into PVDF membrane (100 V, 60 min) in Transfer buffer (Tris-base/Glycine, Sigma Aldrich, Munich, Germany) using the Mini Trans-Blot^®^ system (Bio-Rad, Hercules, CA, USA). After the transfer, the membranes were blocked for 1 h with 5% skim milk (Sigma Aldrich, Munich, Germany). Then, the membranes were incubated overnight at room temperature with primary antibodies detecting osteopontin in dilution of 1:1000 (OPN, ab8448, Abcam, Cambridge, UK), RUNX-2 in dilution of 1:100 (F-2: sc-390351, Santa Cruz Biotechnology, Dallas, Texas, USA), and β-actin in dilution of 1:1000 (A2066, Sigma Aldrich, Munich, Germany). The antibodies were diluted in 5% of skim milk prepared in TBST buffer (Tris/NaCl/Tween). Next, the membranes were washed five times for 5 min with TBST buffer. After washing, the membranes were incubated with secondary antibody conjugated with HRP (Sigma Aldrich, Munich, Germany) for 60 min at room temperature. The secondary antibody was diluted at concentration 1:2500 in TBST. After incubation with secondary antibody the membranes were washed, as indicated above and analyzed using Bio-Rad ChemiDoc™ XRS system. The reaction was performed using DuoLuX^®^ Chemiluminescent and Fluorescent Peroxidase (HRP) Substrate (Vector Laboratories, Peterborough, United Kingdom). The intensity of signals was quantified using Image Lab™ Software (Bio-Rad, Hercules, CA, USA).

### 2.8. Immunocytochemical Detection of Osteopontin and Tartrate-Resistant Acid Phosphatase (TRAP)

Immunostaining reaction was performed according to methods described previously [27,31]. Before the analysis, cells were cultured under experimental conditions, within 24-weel dishes coated with glass cover slides. After the experiment, cultures were fixed with 4% PFA (30 min at room temperature) for the immunocytochemical detection. Following fixation, cultures were washed three times with HBSS and permeabilized with 0.2% PBS-Tween solution for 15 min. Next, specimens were washed 3 times in HBSS and incubated overnight at 4 °C with primary antibodies and 10% goat serum in order to block nonspecific protein–protein interactions. The following antibodies were used: anti-osteopontin antibody produced in rabbit (ab8448, Abcam, Cambridge, UK) and anti-TRAP antibody (D-3) mouse monoclonal IgG1. (sc-376875, Santa Cruz Biotechnology, Dallas, Texas, USA). Primary antibodies were diluted to concentration of 1:50 (TRAP) and 1:1000 (OPN). After incubation with primary antibody, samples were washed as described previously and incubated with secondary antibody for 1 h at room temperature. Concentration of secondary antibodies was 1:1000. After incubation with the secondary antibodies, samples were washed (as above) and fixed on slides using mounting medium with DAPI (4’,6-diamidino-2-phenylindole) as a nuclear counterstain (ProLong™ Diamond Antifade Mountant with DAPI, Thermo Fisher Scientific, Warsaw, Poland). Specimens were analyzed using confocal microscope (Leica TCS SPE, Leica Microsystems, KAWA.SKA Sp. z o.o., Zalesie Gorne, Poland) at 0.5 µm steps up to a final depth of 25 µm. Images were processed using Fiji is just ImageJ (ImageJ 1.52n,Wayne Rasband, National Institute of Health, USA) [27,32]. The microscopic images were obtained using maximum intensity projection (Z-projection).

### 2.9. Statistical Analysis

Experimental values are presented as the mean obtained from at least three technical repetitions. Mean values are presented with standard deviation (±SD). The data were analyzed using t-Student test or One-way analysis of variance and Dunnett’s post hoc test. The data were analyzed using GraphPad Software (Prism 8.20, CA, USA). Differences with a probability of *p* < 0.05 were considered as significant.

## 3. Results

### 3.1. The Influence of miR-21 Inhibition on mRNA Expression of Osteogenic Markers

The expression of osteogenic markers was monitored after 3, 7, and 15 days of differentiation since their profiles are known to be highly modulated during the early stages of osteogenesis Figure 1).

No significant differences in terms of osteocalcin (*Ocl*) levels were noted between MC3T3 cultures and MC3T3*_inh21_* after 3 days of osteogenesis. However, in MC3T3 *_inh21_* co-cultured with 4B12 we noted significant increase of *Ocl* transcripts following 3 days of differentiation (Figure 1a). Significantly reduced mRNA levels for *Ocl*, in response to miR-21 inhibition were noted after 7 and 15 days of MC3T3 differentiation, both in monolayer, as well as in the co-culture system (Figure 1b,c). Further, we determined the mRNA levels for collagen type 1 (*Coll-1*). Analysis revealed that inhibition of miR-21 in MC3T3 cultures do not affect the *Coll-1* expression after 3 and 7 days of differentiation (Figure 1d,e). The significant decrease in mRNA levels for *Coll-1*, following miR-21 inhibition were noted after 15 days of culture (*p* < 0.05, Figure 1f). In co-culture model of MC3T3 with 4B12, inhibition of miR-21 had influence on *Coll-1* levels only after 7 days of differentiation. In these cultures we observed significant (*p* < 0.01) increase of *Coll-1* transcripts in MC3T3 *_inh21_*_/_4B12 cultures. The higher mRNA level for *Coll-1* was also noted in MC3T3 *_inh21_*_/_4B12 cultures following 15 days of osteogenic differentiation, however the observed differences were not statistically significant (Figure 1d,e). The mRNA levels for osteopontin (*Opn*) measured in MC3T3 after 3 days of osteogenic cultures were not affected by miR-21 inhibition (Figure 1g). In turn, after 7 and 15 days of culture we observed significant decrease of *Opn* transcripts (Figure 1h,i). Significantly decreased mRNA levels for *Opn* were determined during 3, 7, and 15 days of osteogenic cultures of MC3T3 *_inh21_* with 4B12 (Figure 1g–i). The inhibition of miR-21 significantly influenced on mRNA levels of runt-related transcription factor 2 (*Runx2*) in MC3T3 cell line. The decrease expression of *Runx-2* following miR-21 inhibition was noted during 3, 7, and 15 days of osteogenesis (Figure 1j–l). In co-culture of MC3T3 with 4B12, the inhibition of miR-21 had no influence on *Runx-2* levels after 3 and 15 days of culture (Figure 1j,l), however lowered expression of miR-21 was correlated with increased mRNA level for *Runx-2*, noted after 7 days of osteogenic differentiation (Figure 1k).

### 3.2. The Analysis of mRNA Transcripts of RANKL-OPG Axis

As the relation between *Rankl* and *Opg* expression levels in osteoblasts determines differentiation of osteoclasts [17], we decided to analyze the influence of the miR-21 inhibition on *Rankl*/*Opg* ratio. Obtained results showed that miR-21 inhibition do not influence on *Rankl*/*Opg* ratio during 3 and 7 days of culture under osteogenic conditions (Figure 2a and b). Significantly increased *Rankl* levels (*p* < 0.05) were noted in MC3T3*_inh21_* cultures after 15 days of osteogenesis (Figure 2c).

In turn we observed decreased expression of *Rankl* in co-cultures of MC3T3*_inh21_* with 4B12. Decreased *Rankl*/*Opg* ratio in MC3T3*_inh21_/*4B12 was maintained during the osteogenesis, on days 3, 7, and 15 (Figure 2a-c).

### 3.3. The Influence of miR-21 Inhibition on Extracellular Matrix Composition

Calcium deposition in extracellular matrix is a marker of late osteogenesis and reflects maturation of osteoblasts precursors [1,11,33], thus we decided to determine the extracellular matrix composition after 15 days of osteogenesis. Largely distributed calcium deposits were detected in pre-osteoblasts (M3CT3) maintained under osteogenic conditions, indicating that cells properly underwent the differentiation process (Figure 3a). The inhibition of miR-21 caused a decrease in extracellular matrix mineralization. The MC3T3*_inh21_* cells formed dense networks, however we did not observe the formation of characteristic nodules stained with Alizarin Red (Figure 3b). Furthermore, the osteoclasts presence significantly lowered osteogenic differentiation of MC3T3. In co-culture of MC3T3 with 4B12 we observed a decreased number of mineralized areas (Figure 3c). In turn, the production of extracellular calcium deposits in co-cultures of MC3T3*_inh21_* with 4B12 was more apparent (Figure 3d). Spectrophotometric analysis of dye absorption and statistical analysis of the obtained results, confirmed the observations made under microscope. SEM-EDX analysis confirmed lowered calcium and phosphorous deposition in MC3T3 cultures with decreased expression of miR-21 and in co-culture model (Appendix A).

We also determined the influence of miR-21 inhibition on proteoglycan content in extracellular matrix formed by the cells under osteogenic conditions (d15). Images obtained after Safranin-O staining (Figure 4) coincide with the results of Alizarin Red analysis (Figure 2). The most abundant accumulation of proteoglycans was observed in osteogenic cultures of MC3T3 (Figure 4a). Transfection of pre-osteoblasts with the miR-21 inhibitor, as well as co-culture of MC3T with 4B12 pre-osteoclasts diminished the synthesis of proteoglycans and their deposition in the extracellular matrix (Figure 4b,c). The intense reaction indicating high proteoglycan content was noted in MC3T3*_inh21_* co-cultured with pre-osteoclasts (Figure 4d). Comparative analysis of staining intensity confirmed microscopic observations (Figure 4e).

### 3.4. The Influence of miR-21 Inhibition on Intracellular Accumulation of OPN and RUNX-2 in MC3T3 after 15 Days of Osteogenesis

Western blot analysis was performed to determine the influence of the miR-21 inhibition on intracellular expression of OPN and RUNX-2 in pre-osteoblast maintained under osteogenic conditions. The analysis indicated the presence of three different immunoreactive bands for OPN having various molecular weight: 66 kDa resembling full-length protein, as well as 30 kDa and 24 kDa bands resulting from the full-length protein cleavage (Figure 5a). The densitometry analysis of band intensities indicated that the inhibition of miRNA-21 decreases the expression of 66 kDa OPN, however the differences were not statistically significant (Figure 5b). Furthermore, the presence of osteoclasts precursors in osteogenic cultures of MC3T3 (co-cultures) resulted in significant accumulation of 66 kDA OPN. Next, we analyzed the levels of 30 kDa OPN, and we observed that the band intensity is not altered by miR-21 inhibition for cells cultured in the monolayer system (Figure 4c). In the co-culture of MC3T3 with 4B12 we noted significant increase of 30 kDa protein. In contrast to that, we determined the significant decrease of protein in homogenates derived from MC3T3*_inh21_* cultured with osteoclasts. The expression of 25 kDa OPN was notably increased in the osteogenic cultures of MC3T3, the protein level significantly decreased after miR-21 inhibition also in the presence of osteoclasts (Figure 5d). The expression of 65 kDa and 55 kDA RUNX-2 decreased after miR-21 inhibition (Figure 5e,f), however significant difference in terms of signal intensity was noted only for 65 kDa bands. The expression of RUNX-2 decreased in the co-culture system of MC3T3/4B12. The inhibition of miR-21 expression in MC3T3 cultured with pre-osteoclast had no effect on RUNX-2 intracellular accumulation.

Immunocytochemistry was used to determine the OPN cellular localization (Figure 6). Images obtained with confocal microscope confirmed that OPN expression decreases after the miR-21 inhibition. Moreover, the increase of OPN expression was confirmed in MC3T3/4B12 cultures. The visualization of actin cytoskeleton indicated that under osteogenic conditions, MC3T3 are characterized by well-developed network supporting intracellular connections. The inhibition of miR-21 caused the alteration in cytoskeleton organization by weakening the architecture of actin network. In the co-culture system of MC3T3 and 4B12, the loss of intracellular connections was noted. Images obtained for MC3T3_inh21_/4B12 co-cultures indicated an increase in number of osteoblasts and maintenance of intracellular connections between pre-osteoblast.

### 3.5. The Analysis of Markers Associated with Differentiation and Bone Resorption Activity of Osteoclasts

Bearing in mind the *Rankl*/*Opg* profile expression obtained for MC3T3 cultures after 15 days of osteogenesis, we were interested in an analysis of markers that are strictly characteristic for the activity of osteoclast cells, as well as for their survival. The analysis included measurement of mRNA levels for receptor activator of nuclear factor κB (*Rank)*, tartrate-resistant acid phosphatase (*Trap*), cathepsin K (*Ctsk*), carbonic anhydrase II (*CaII*), and matrix metalloproteinase 9 (*Mmp-9*), as well as genes from B-cell lymphoma 2 family associated with apoptosis, i.e., pro-apoptotic *Bax* and anti-apoptotic *Bcl-2*. Obtained results showed significantly increased levels of transcripts for *Rank*, *Trap*, *Ctsk*, *CaII*, and *Mmp-9* in 4B12 co-cultured with MC3T3 without inhibitor (Figure 7a–d) which indicates increased pre-osteoclasts activity. In contrast, the osteoclasts cultured with MC3T3_inh21_ expressed pro-apoptotic profile related to increased levels of mRNA for *Bax* and decreased for *Bcl-2* (Figure 7e).

Following that, TRAP expression was studied using immunocytochemistry technique. We did not observe a specific signal derived from TRAP positive cells in MC3T3 and MC3T3*_inh21_* cultures (Appendix A). In turn, the analysis of the TRAP expression performed on co-cultures of pre-osteoblast and 4B12 revealed that occurrence of TRAP positive cells increased in the MC3T3/4B12 co-culture system, while decreasing in MC3T3*_inh21_*/4B12 (Figure 8), which confirms the results obtained by RT-qPCR (Figure 7b).

## 4. Discussion

The number of studies related to function of miR-21 in bone homeostasis maintenance has grown rapidly over the past few years. Despite the efforts, the function of miR-21 in the biology of bone forming and bone-resorbing cells has not been fully elucidated. The aim of this study was to investigate the role of miR-21 during differentiation of osteoblast precursors—MC3T3 cell line. In addition, we determined the effect of miR-21 inhibition on osteoblast interaction with osteoclasts precursors using indirect co-culture system of MC3T3*_inh21_* and semi-adherent pre-osteoclasts cell line 4B12 [18]. This model allowed us to study paracrine interactions between differentiating osteoblasts and osteoclasts [34]. We used this strategy to monitor individual phenotype changes and gene expression characteristics, both in osteoblast and osteoclast, which is difficult in the condition of direct co-culturing. Moreover, the indirect co-culture model is more accurate for cells with different characteristics of growth [35,36].

Various research groups put forward compelling evidence that miR-21 promotes differentiation of progenitor cells toward osteoblasts. The function of miR-21 as an essential molecule promoting bone regeneration was determined for example using model of multipotent stromal cells (MSCs) [2,7,12,14]. However, discussed results were inconsistent and contradictory, indicating that miR-21 may promote, as well as inhibit osteogenic differentiation, what can be correlated with diverse cellular plasticity of MSCs and their origin. To minimize this issue, we used the model of pre-osteoblast MC3T3 cell line that represents a stable and reproducible model for studies related with signaling pathways crucial for proliferation and differentiation of osteoblast [37,38,39,40,41]. Furthermore, under osteogenic conditions MC3T3 pre-osteoblasts synthesize and assemble collagenous extracellular matrix with organization and mineralization that resembles bone [37]. Previously, it was shown that overexpression of miR-21 in MC3T3 pre-osteoblast improves matrix mineralization, whereas the miR-21 inhibition was correlated with lowered calcium deposition [11]. The results are in good agreement with our data, confirming that miR-21 inhibition is associated with decreased deposition of calcium and lowered proteoglycan content in extracellular matrix (ECM).

We have also noted reduced mineralization of ECM in the co-culture model of MC3T3/4B12, which correlates with high occurrence of TRAP positive cells and increased mRNA levels for *Rank*, and other osteoclastic bone resorption-related genes like MMP-9, cathepsin K (*Ctsk*) and carbonic anhydrase II (*CaII*). In turn, the activity of TRAP-positive cells was reduced in co-cultures of MC3T3_inh21_/4B12, indicating that inhibition of miR-21 in osteoblast may reduce paracrine signals necessary for pre-osteoclasts survival [42]. Thus, the mineralization of ECM in MC3T3_inh21_/4B12 culture was less influenced by osteoclasts activity.

As miR-21 regulates the process of osteogenesis at the molecular level, we decided to determine the mRNA expression of main osteogenic markers after 3, 7, and 15 days of osteogenic stimulation. We showed that miR-21 inhibition is associated with decreased mRNA levels for collagen I (*Coll-1*), osteocalcin (*Ocl*) and *Runx-2*, which was previously reported by Li et al. [11]. We also studied the influence of miR-21 inhibition on the mRNA levels of osteopontin (*Opn*). This molecule plays a crucial role in bone remodeling by influencing differentiation of osteoblasts and determining survival of osteoclasts. An effect of miR-21 overexpression on *Opn* mRNA levels was recently studied using BMSCs model [7]. Authors showed that *Opn* expression increases in BMSCs as a result of miR-21 upregulation [7]. Our results confirmed the role of miR-21 as a molecule regulating *Opn* expression. MC3T3 with downregulated expression of miR-21 had lowered *Opn* expression, measured at 7 and 15 days of in vitro osteogenesis. Nevertheless, we observed that *Opn* mRNA level increases in the co-culture of MC3T3 with 4B12 cells. This is in agreement with studies showing that osteopontin plays crucial role in modulating osteoclast-osteoblast interactions, both in vivo and in vitro. Boskey et al. [43] showed that OPN knockout mice are characterized by elevated mineral content and crystallinity in bone, while OPN knockout osteoblasts formed matrix with increased mineral deposition [44]. This observation may explain the improvement of matrix mineralization in model of MC3T3*_inh21_* propagated with 4B12 pre-osteoblasts.

Having the results obtained on mRNA level, we decided to determine the expression of OPN using Western blot technique. We detected three different immunoreactive bands of OPN, i.e., 66 kDa, as well as 30 and 25 kDa. We observed that in the co-culture system the expression of OPN at molecular weight equal to 66 kDa (doublets) increases, in contrast to the monolayer cultures. The miR-21 inhibition does not have any influence on OPN expression at 66 kDa. However, in MC3T3/4B12 cultures we observed the increased expression of OPN at 35 kDa, and its levels significantly decreased in the co-cultures of MC3T3*_inh21_*/4B12. Obtained results confirm previous data, indicating that OPN produced by osteoblasts can activate bone resorption by osteoclasts [45]. The OPN expression after miR-21 inhibition may explain the images obtained after ECM staining, showing the improvement in matrix biomineralization in MC3T3*_inh21_*/4B12 cultures. Osteogenic cultures of MC3T3 were characterized by increased expression of smaller molecular-weight OPN protein—25 kDa. The function of low molecular weight OPN is not properly described in the literature, however it was shown that this protein may be a product of OPN cleavage with matrix metalloproteinases [46]. We also determined the intracellular localization of OPN in the experimental cultures, and the results of analysis were consistent with data obtained using RT-qPCR, showing decrease of OPN after miR-21 inhibition in MC3T3 cells as well as increase of OPN expression in the co-cultures of MC3T3 with 4B12 pre-osteoclasts.

Further, we confirmed that the miR-21 inhibition may significantly reduce the intracellular accumulation of RUNX-2, which is a common regulator of osteogenesis [47,48]. The obtained results are consistent with recent studies showing that BMSCs after transfection with the miR-21 inhibitor exhibit lowered accumulation of intracellular RUNX-2 on both mRNA [1] and protein level [12].

A study performed by Li et al. indicated that miR-21 regulates differentiation of progenitor cells into bone forming cells via the Smad7-Smad1/5/8-Runx2 pathway. Moreover, it was shown that miR-21 deficiency impaired the bone formation of calvarial bone defects, what might result from deterioration of osteogenic potential of endogenous progenitor cells. Multipotent stromal cells derived from bone marrow of miR-21 knock-out mice, exhibited lowered osteogenic potential associated with decreased mRNA levels for Runx-2 [12]. Additionally, Meng et al. showed that miR-21 promotes osteogenic differentiation of human umbilical cord mesenchymal stem cells (hUMSCs) through PI3K/β-catenin pathway, activating the transcription of RUNX-2 [14]. However expression of *Runx-2* is modulated at multiply levels during the osteogenesis. Runx-2 expression profile strictly depends on differentiation stage of the cells. It was previously indicated that Runx-2 may act as a positive regulator during early stages of osteoblast differentiation and negative regulator at later stage of osteogenesis [47,48,49]. Given the stage-dependent shift of Runx2 expression, the role of miRNA-21 as a regulator of Runx-2 should be monitored at early, mid and late stage of osteogenesis, what was also confirmed in our study.

It was speculated previously, that RUNX-2 controls RANKL expression and that this interaction conveys molecular background for the linkage between osteoblast and osteoclast formation [50]. Our results confirmed this dependency. We have shown that RANKL/OPG ratio is increased in MC3T3 pre-osteoblasts transfected with the miR-21 inhibitor, thus we expected the increased osteoclast formation and activity in this condition. However, in the co-culture system of MC3T3_inh21_/4B12 we observed the decrease of RANKL/OPG ratio. Obtained result confirmed the study of Pitari et al. who showed that co-culture of multiple myeloma cells with BMSCs under constitutive inhibition of miR-21 may result in the restoration of RANKL/OPG balance and impair the resorbing activity of mature osteoclasts [51]. This implies, that the miR-21 inhibition may affect the activity of pre-osteoclasts and their survival regulating RANKL expression. This is in agreement with studies of Sutherland et al. [52], showing that RANKL protects osteoclasts from the apoptosis-inducing and anti-resorptive effects of bisphosphonates in vitro, via anti-apoptotic protein Mcl-1. In turn, in our study we showed that the inhibition of miR-21 in pre-osteoblast is associated with the lowered expression of Rankl, and increased Bax/Bcl-2 ratio. Obtained results coincide with research of Hu et al. showing that pro-osteoclastic effect of miR-21 may be associated with its function during regulation of programmed cell death [53]. Moreover, the most recent research of Luukkonen et al. [54] showed that the inhibition of miR-21 associated with decreased OPN accumulation affects survival of pre-osteoclasts. The OPN is essential for osteoclasts attachment to the resorbed matrix and the cell surface receptor. Luukkonen et al. showed that OPN produced in the resorption pits may inhibit bone mineralization, what also confirms the suppressive effect of OPN on osteoblast physiology emerging from autocrine/paracrine signals. Of note, confocal analysis of MC3T3_inh21_/4B12 co-cultures showed decreased TRAP expression which was also evidenced at mRNA level. This result is in line with study of Ek-Rylander and Andersson, who indicated that TRAP could regulate the extent of ECM degradation including depth and area at each bone resorption site [55]. The study showed that TRAP modulates OPN-dependent osteoclasts migration and triggers osteoclast detachment facilitating their subsequent movement on the bone surface. Given the increased apoptosis of pre-osteoclasts cultured with MC3T3*_inh21_*, we hypothesize that lowered viability of TRAP positive pre-osteoclasts could be solely due to altered osteoblast signaling.

## 5. Conclusions

In summary, we demonstrated that decreased expression of miR-21 in MC3T3 pre-osteoblasts cell line results in loss of their bone forming capability, what was evidenced by poor mineralization of extracellular matrix in vitro and lowered expression of essential bone markers. Moreover, the inhibition of miR-21 expression attenuated paracrine activity of pre-osteoblast, which was associated with increased apoptosis of pre-osteoclasts in the co-culture model. The observed effect may be related with the function of miR-21 as a regulator of key molecules regulating osteoblastogenesis and osteoclastogenesis. MiR-21 may have a dual role in the process of osteoblast-osteoclast coupling, probably due to the fact that its targets (eg. *Rankl* and *Opn*) are regulated in a dynamic manner during the process of osteogenesis. Thus, analysis of regulatory effect of miR-21 should be determined at different stages of osteogenesis in order to elucidate its influence on genes expressed during early osteogenic commitment (d3, d7, and d15). Obtained results are in line with other studies indicating the emerging role of miR-21 in terms of regulation of bone metabolism and homeostasis. Profound characterization of miR-21 function in osteoblast-osteoclast coupling may account for development of novel therapies directly influencing bone cell differentiation and the process of bone remodeling.

## Figures and Tables

**Figure 1 cells-09-00479-f001:**
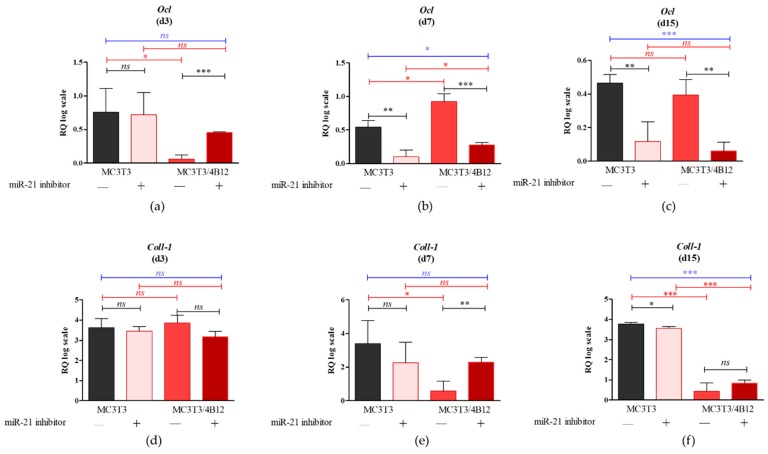
The results of RT-qPCR analysis showing the mRNA levels of key osteogenic markers measured after 3 (**a**,**d**,**g**,**j**), 7 (**b**,**e**,**h**,**k**), and 15 days (**c**,**f**,**i**,**l**) of osteogenic cultures (d3 d7 and d15 respectively). Following osteogenic markers were determined: osteocalcin (*Ocl*) (**a**–**c**), collagen type I (*Coll-1*) (**d**–**f**), osteopontin (*Opn*) (**g**–**i**) and runt-related transcription factor 2 (*Runx2*) (**j**–**l**). Significant differences are indicated with asterisks (* *p* < 0.05; ** *p* < 0.01 and *** *p* < 0.001), while non-significant differences are marked as *ns*. The comparisons between groups are marked with square brackets.

**Figure 2 cells-09-00479-f002:**
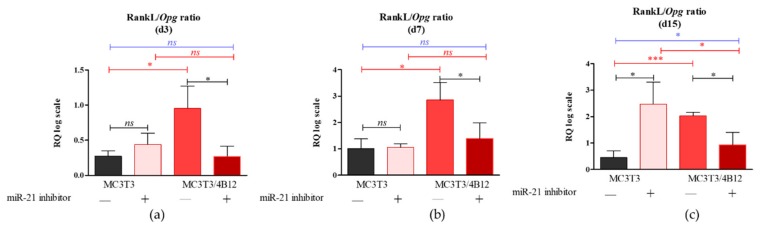
The *Rankl/Opg* ratio determined in experimental osteogenic cultures on days 3 (**a**), 7 (**b**), and 15 (**c**). Significant differences are indicated with asterisks (* *p* < 0.05 and *** *p* < 0.001), while non-significant differences are marked as *ns*. The comparisons between groups are marked with square brackets.

**Figure 3 cells-09-00479-f003:**
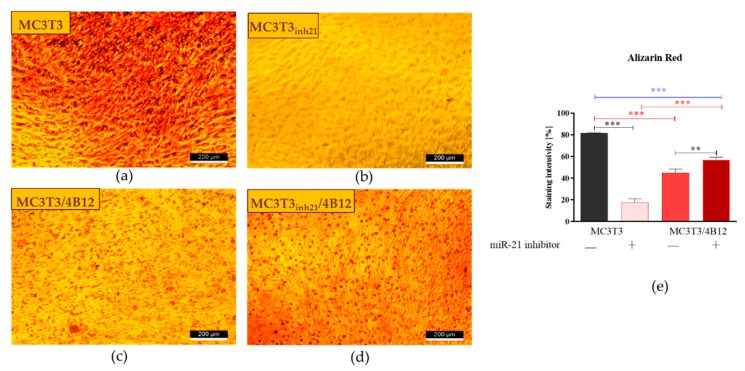
Representative images showing the results of Alizarin Red staining for calcium deposit detection. The mineralized extracellular matrix is stained with the dye. The images show an effect of miR-21 inhibition on mineralization of extracellular matrix formed by pre-osteoblast. The analysis included the determination of osteoclast precursors’ activity (co-culturing with 4B12) on MC3T3 osteogenic differentiation. Osteogenic differentiation of MC3T3 was manifested by red calcium deposits (**a**). The decreased mineralization was observed in cultures of MC3T3 with blocked activity of miR-21 (**b**). The influence of 4B12 pre-osteoclast presence during osteogenic differentiation was monitored both in co-culture with MC3T3 (**c**) and MC3T3*_inh21_* (**d**). Images were taken under 100-fold magnification (scale bar indicated). Staining intensity was determined using ImageJ software with Pixel Counter application, as indicated in Materials and Methods section (**e**). Significant differences are marked using an asterisks (** *p* < 0.01 and *** *p* < 0.001), while non-significant differences are described as *ns*. The comparisons between groups are marked with square brackets.

**Figure 4 cells-09-00479-f004:**
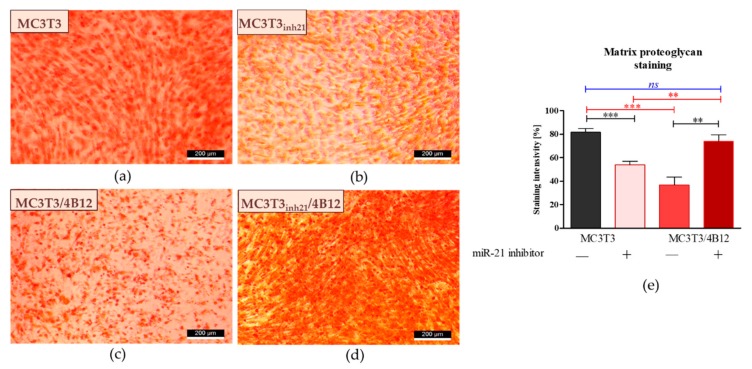
Representative images showing detection of proteoglycans with Safranin-O. The images show the effect of miR-21 inhibition on proteoglycan accumulation within the extracellular matrix formed by pre-osteoblasts under osteogenic conditions. Additionally, influence of osteoclast precursors on MC3T3 differentiation was evaluated. Osteogenic differentiation of MC3T3 was associated with increased accumulation of proteoglycans (**a**). The decreased content of proteoglycans was observed in cultures of MC3T3 transfected with miR-21 inhibitor (**b**). The influence of 4B12 pre-osteoclast presence during osteogenic differentiation of MC3T3 was monitored. The analysis revealed that osteoclasts presence decreased the accumulation of proteoglycans (**c**). In turn, extracellular matrix which formed in co-cultures of MC3T3_inh21_ with osteoclasts was rich in proteoglycans (**d**). Images were taken under 100-fold magnification (scale bar indicated). The staining intensity was determined using ImageJ software with Pixel Counter application, as indicated in Materials and Methods section (**e**). Significant differences are indicated with asterisks (** *p* < 0.01 and *** *p* < 0.001), while non-significant differences are marked as *ns*. The comparisons between groups are marked with square brackets.

**Figure 5 cells-09-00479-f005:**
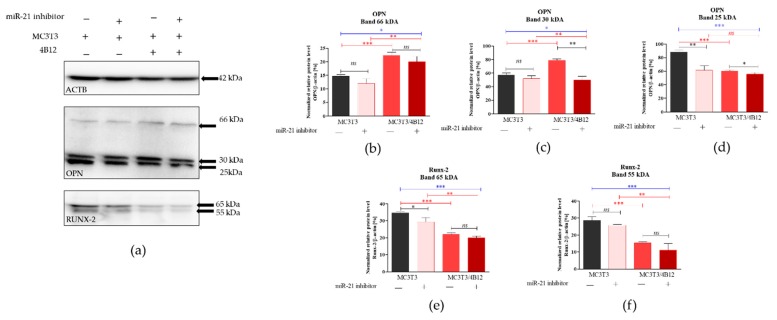
Intracellular accumulation of osteopontin (OPN) and runt-related transcription factor 2 (*Runx-2*). The β-actin (ACTB) was used as a housekeeping protein for normalization. The representative blots are shown in graph (**a**) and molecular weights of the detected proteins are indicated on the right. Results of statistical analysis performed on normalized values are shown in graphs (**b**–**f**). Significant differences are indicated with asterisks (* *p* < 0.05; ** *p* < 0.01 and *** *p* < 0.001), while non-significant differences are marked as *ns*. The comparisons between groups are marked with square brackets.

**Figure 6 cells-09-00479-f006:**
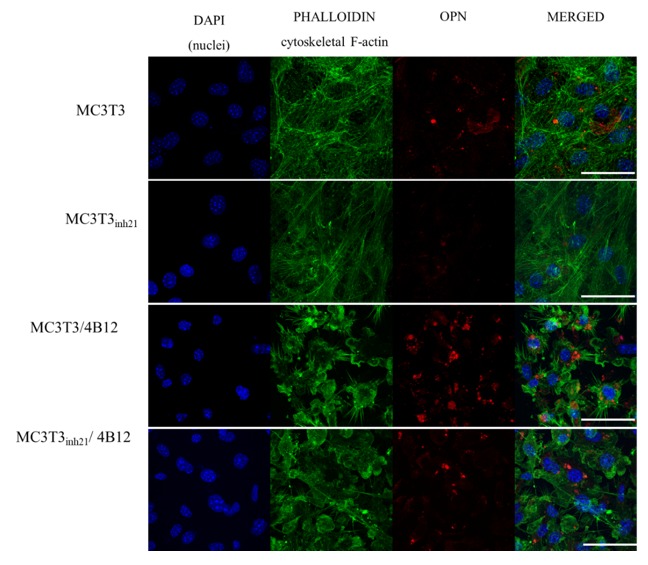
Representative images (Z-projects) showing co-localization of OPN (red signal) with nuclei (blue, DAPI stained) and cytoskeleton (green, phalloidin atto-488 stained). The images were taken under 60-fold magnification. The scale bar is equal to 50 μm.

**Figure 7 cells-09-00479-f007:**
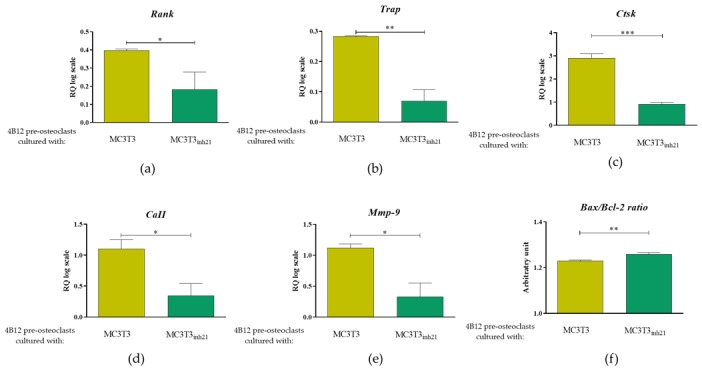
The expression of mRNA for typical osteoclast markers and molecules associated with apoptosis. The analysis included determination of transcript levels for receptor activator of nuclear factor κB (**a**), tartrate-resistant acid phosphatase (**b**), cathepsin K (**c**), carbonic anhydrase II (**d**) and matrix metalloproteinase 9 (**e**). Moreover, the ratio of pro-apoptotic *Bax* and anti-apoptotic *Bcl-2* gene was calculated (**f**). Significant differences between groups are indicated with asterisks (* *p* < 0.05; ** *p* < 0.01 and *** *p* < 0.001), while non-significant differences are marked as *ns*.

**Figure 8 cells-09-00479-f008:**
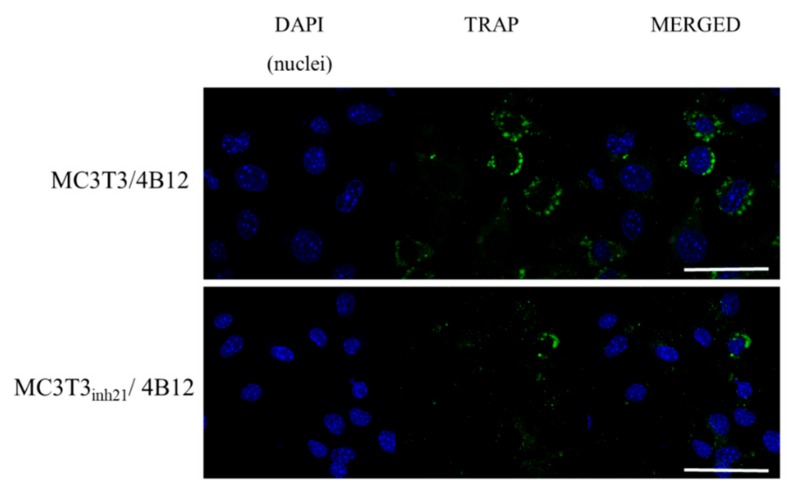
Representative images showing co-localization of tartrate-resistant acid phosphatase (TRAP) (green signal) with nuclei (blue, DAPI stained). The images were taken under 60-fold magnification. The scale bar is equal to 50 μm.

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
