# Peer review of "The Role of miR-21 in Osteoblasts–Osteoclasts Coupling In Vitro"

_cells, 2020, doi:10.3390/cells9020479_

Round 1
Reviewer 1 Report
Manuscript # 694183 by Agnieszka Smieszek et al. describes an in vitro study based on an indirect co-culture method of murine pre-osteoblastic and pre-osteoclastic cell lines. The preosteoblastic cell line was treated with a miR-21 inhibitor; the authors’data demonstrate how miR-21 inhibition not only impairs osteogenesis, but it also affects osteoclastogenesis and osteoclast survival.
Comments:
the text is misspelled and definitely needs revision by a native English speaker. Experimental data are rather accurate, nevertheless I wish to point out the following criticisms:a)-In order to exclude any effects due to the transfection procedure, a negative control (using scrambled sequence) should be added. Please report these data
b)-It seems that all analyses in MC3T3 cells were performed after 15 days cultivation in osteogenic medium. Osteogenic markers are highly modulated during the differentiation process. In particular, it is important to evaluate RUNX2 expression levels after 3 and 7 days of differentiation. The effects of miR21 absence/presence may differ during the various phases of the differentiation process.
Overall evaluation:
The interesting point of this work, in my opinion, is the indirect co-culture approach, where miR-21 inhibition was obtained in pre-osteoblastic cells. The data presented confirm how such inhibition affects their paracrine pro-osteoclastogenic activity. These findings do notrepresent a novelty, anyhow, since the complex role of miR-21 in bone metabolism, being a pro-osteogenic as well as pro-osteoclastic microRNA, has been investigated extensively in KO mice ( Hu CH et al., Sci Rep 2017) and in human cells (Pitari R et al. Oncotarget 2015). [Both references have been cited in the manuscript under review].
The quality of this manuscript could be improved, as suggested above.
Reviewer 2 Report
The authors studied the effect of miR-21 on pre-osteoblasts and pre-osteoclasts coupling in vitro. Specific points are as follow.
Abstract (line 28)- “….miR-21 can be vital factor for osteoblasts to promote survival of pre-osteoclast.” The authors should revise the statement as osteoblasts shown too produce RANKL and M-CSF to promote pre-osteoclast survival.
Introduction (line 103)- “… osteoclast precursor cell line i.e. 4B12.” Remove the word ‘novel’ as citation by Amano group as noted (ref18) is published in 2009.
Methods (p.4, line 153) clarify the sentence as “……..4B12 was kindly provided by Shigeru…”
Results (p.10) subtitle “3.5. The analysis of markers associated with osteoclasts survival”-pleas rephrase the subtitle as the gene expression analyzed is more of osteoclast differentiation/bone resorption activity than survival.
Fig.8- (p.11, line 391) “… TRAP expression was studied in co-cultures of pre-osteoblast and 4B12 using immunocytochemistry technique. Absence of TRAP expression in MC3T3 cells is not identified. The authors should explain why they prefer immunocytochemistry over cytochemical staining or western blot analysis for protein expression.
Discussion- RANKL is reported as osteoclast survival factor. if miR-21 deficiency is increases RANKL expression, the authors should discuss the implications appropriately.
Round 2
Reviewer 1 Report
to Śmieszek et al. from reviewer 1:
The English style still needs polishing. (e.g. we have showed at line 509 should be: we have shown; generally speaking, in the revised version an excessive and not always appropriate utilization of definite article “the” has been introduced). Not being an English mother tongue person, I do not feel entitled to correct the whole manuscript, but a native English speaker should do it.
Your response to query a) is acceptable
Your response to query b) is not acceptable.
Your statement “at the time point selected (15 days cultivation in osteogenic medium) the highest expression of osteogenic markers is observed, including RUNX2” is not correct. Many studies contradict it, I just want to mention the following reference, where the authors investigated MC3T3 cell line, as well as you did: Preziger S. et al. PROGRESSIVE RECRUITMENT OF RUNX2 TO GENOMIC TARGETS DESPITE DECREASING EXPRESSION DURING OSTEOBLAST DIFFERENTIATION. J Cell Biochem 2008;105:965-970.
I think you should mention, in the discussion, this limit in your experimental approach( not having monitored RUNX2 expression modulation at earlier stages of osteogenic differentiation) and therefore you can not affirm that what you measured was the highest expression level of RUNX2 expression.
Author Response
Please see the attachment.

This manuscript is a resubmission of an earlier submission. The following is a list of the peer review reports and author responses from that submission.
Round 1
Reviewer 1 Report
An interesting and currently relevant piece of work regarding the effect of miR-21 inhibition on osteoblasts. I do have some suggestions for improvement/requests for clarification:
1) As a general point, the grammar was poor throughout (even the title was problematic).
2) Methods - your information regarding cell maintenance and passaging is insufficient. How were osteoclasts received/maintained prior to use? How many passages did your cells undergo? Exactly what trypsin did you use (trypsinization can have a significant effect on osteoblastic cells)?
If all of the work was in vitro, why were mouse cells used, as opposed to human?
3) Results -
a) In figures 1 and 2, the graph data does not appear to accurately reflect the images. In 1, for e.g., the pics show clearly a>d>c>b, but your graphs show a>b&d>c. You could perhaps display the actual absorbance values. I would assume all the data from each graph represents the same control values etc. between groups (otherwise it would be misleading)?
b) In figure 4, a) and b) do not seem to reflect each other. Also, is the control shown in a) the control for all blots shown (i.e. they were all taken from the same membrane)?
c) In figure 5, were the images shown z-stacks, or only one plane? It is important you provide the image analysis information. For example, the differences in actin observed could be due to the images being obtained at different cell heights.
Discussion - it should be made clear that the differences in osteoclasts could be due solely to altered osteoblast signalling.
Some key issues:
No evidence was provided showing the direct interaction of miR21 with the shown targets (e.g.OPN). It is therefore unclear whether the targets shown directly interact with miR21, or there is a factor upstream that the authors did not identify.This should be addressed.
A control of osteoblasts where differentiation was inhibited by an agent rather than miR-21 should have been utilized, as should a control miRNA.
Reviewer 2 Report
This manuscript reports on the potential function of miR-21 as a regulatory molecule in the coupling of osteoblasts and osteoclasts. The authors used an in vitro approach utilizing a miR-21 inhibitor to demonstrate the potential function of this regulatory RNA in affecting osteoblast differentiation and osteoclasts as well. Although the manuscript is thorough and presents convincing data, it needs some revisions as outlined below that could strengthened it before it can be accepted in Cells.
Document needs some English editing; sentence structures, missing periods, spacing between sentences, tenses; etc. I recommend that the authors fix the title to read: “The role of miR-21 in osteoblasts-osteoclasts coupling in vitro”. It is not very clear as it is written now. Lines 120 and 134, please give the exact number of cells. Table S1 should include the melting temperature for each gene. Section on immunoblotting (lines 202-221), should give the concentration of each primary antibody, not just the secondary. Lines 217-218 are not necessary since the previous sentences talk about DAPI staining. The description of results is not adequate; I encourage the authors to include more descriptive details that are quantitative in nature. Simply stating “significantly lower” or “greater” is not good enough. Be specific in your description, i.e. by how much was significantly lower or by how much was it greater, etc. Clearly, some the differences between the control and experimental conditions were extremely small (i.e. Fig. 1e, first 2 bars; Fig. 3b and d, first 2 bars; Fig. 4e, first 2 bars, Fig. 7c). Without knowing the exact difference, it is hard to make a more accurate conclusion about the data. Did the authors investigate cell proliferation in the presence or absence of inhibitor with the two cell lines? If yes, what were the results? I am also surprised that overexpression of mir-21 was not conducted. Why? Figure 3b, I am surprised that Collagen type I levels were only slightly decreased given the large decreases observed with Ocl and Opn. The authors should offer an explanation. Figure 4, why only look at two of the genes from Figure 3? Why not all four for consistency? Perhaps the authors can offer an explanation. Figure 5, lines 317-319, it is not very clear from the images shown in figure that the cytoskeleton in MC3T3ihb21 change as much in comparison to MC3T3. The authors should provide higher magnification images where the actin filaments are more clearly visible.I am surprised that only one marker for osteoclast proliferation was used (Trap). It would be good to see the expression of other classical osteoclast markers (Cathepsin K, Stamp, etc.).
Reviewer 3 Report
Recent studies implied the significance of miR-21 in bone homeostasis/metabolism in vitro and in vivo. Based on this fact, this manuscript by Smieszek et al., tried to reveal the role of miR-21 on pre-osteoblasts activity and regulatory function of osteoclastgenesis. As a result, they concluded that miR-21 facilitates pre-osteoblasts bone forming activity. In addition, miR-21 is also essential factor for osteoblasts to induce osteoclastgenesis. This concept should be novel and interesting.
However, unfortunately, the manuscript is poorly written to interpret their data and technically some experiments is not sound and solid. Thus, it is dubious as to whether as such the work is suitable for Cells.
Major Concerns:
In the Materials and Methods section, there is no information how the authors obtained pre-osteoclast cell line 4B12.
Since the authors did not use gene manipulation system for osteoclasts culture, primary macrophage but not 4B12 could be suitable for osteoclastgenesis assay in vitro.
In general, osteoclastgenesis should be assessed by the number of multi nuclear gigantic cells that express TRAP. However, their 4B12 did not show such differentiation. This is definitely unacceptable for osteoclastgenesis analysis.
The authors employed indirect co-curlture system by using pre-osteoblasts and pre-osteoclasts. This is one of the lethal problems in this manuscript. Direct co-culture is essential for their experiments.
For figure 1 and 2, macroscopic findings are needed. Moreover, to quantify the mineral deposition, they have to measure the amount of Alizarin red in each tested group.
Their qPCR, Western blotting, and immunofluorescence data in figure 3, 4, and 5 are not reliable. For instance, qPCR analysis demonstrated that inhibition of miR-21 drastically reduced OPN mRNA expression, though its protein production was not changed in their western blotting. Moreover, incongruously, immunofluorescence analysis demonstrated that OPN protein production was decreased by miR-21 in MC3T3 cells. Taken together, their results could not support each other and be hard to secure the reliability of their experimental techniques. Same things were also true for RUNX2 expression in figure 3 and 4.
To detect OPN in western blotting and immunofluorescence analysis, the authors used same antibody which can bind to both cleaved and no-cleaved OPN proteins. However, they clearly stated that their confocal images indicate intracellularly accumulated 25kDa OPN. The reviewer cannot understand this theory.
The authors described that co-culture system of MC3T3 and 4B12 decreased the number of osteoblasts. However, the number of DAPI in their immunofluorescence images were obviously comparable among MC3T3, MC3T3inh21, and MC3T3/4B12. In addition co-culture with MC3T3inh21 and 4B12 apparently increased the number of osteoblasts.
Generally speaking, it is hard to understand why they conducted indirect co-culture system for figure 1 to 6. One of the main theme in this study is to investigate whether miR-21 plays a role in osteoblasts-induced osteoclastgenesis. However, the data obtained from their co-culture study indicate the effect of trophic factors provided from 4B12 cells on osteoblasts differentiation.
Round 2
Reviewer 3 Report
The reviewer does not have any more comments.
Although the authors revised their manuscript, the quality seems to be still low level and the manuscript does not deserve acceptance. However, reading their response letter, I think they will not conduct any additional experiments to address my concerns.